# Fast-Evolving Alignment Sites Are Highly Informative for Reconstructions of Deep Tree of Life Phylogenies

**DOI:** 10.3390/microorganisms11102499

**Published:** 2023-10-05

**Authors:** L. Thibério Rangel, Gregory P. Fournier

**Affiliations:** Department of Earth, Atmospheric, & Planetary Sciences, Massachusetts Institute of Technology, Cambridge, MA 02139, USA; g4nier@mit.edu

**Keywords:** phylogeny, multiple sequence alignment, site-specific substitution rate, Tree of Life

## Abstract

The trimming of fast-evolving sites, often known as “slow–fast” analysis, is broadly used in microbial phylogenetic reconstruction under the assumption that fast-evolving sites do not retain an accurate phylogenetic signal due to substitution saturation. Therefore, removing sites that have experienced multiple substitutions would improve the signal-to-noise ratio in phylogenetic analyses, with the remaining slower-evolving sites preserving a more reliable record of evolutionary relationships. Here, we show that, contrary to this assumption, even the fastest-evolving sites present in the conserved proteins often used in Tree of Life studies contain reliable and valuable phylogenetic information, and that the trimming of such sites can negatively impact the accuracy of phylogenetic reconstruction. Simulated alignments modeled after ribosomal protein datasets used in Tree of Life studies consistently show that slow-evolving sites are less likely to recover true bipartitions than even the fastest-evolving sites. Furthermore, site-specific substitution rates are positively correlated with the frequency of accurately recovered short-branched bipartitions, as slowly evolving sites are less likely to have experienced substitutions along these intervals. Using published Tree of Life sequence alignment datasets, we also show that both slow- and fast-evolving sites contain similarly inconsistent phylogenetic signals, and that, for fast-evolving sites, this inconsistency can be attributed to poor alignment quality. Furthermore, trimming fast sites, slow sites, or both is shown to have a substantial impact on phylogenetic reconstruction across multiple evolutionary models. This is perhaps most evident in the resulting placements of the Eukarya and Asgardarchaeota groups, which are especially sensitive to the implementation of different trimming schemes.

## 1. Introduction

The suitability of alignment sites for use in phylogenetic inference is influenced by many factors, such as compositional bias, site-specific substitution frequencies, and evolutionary rates [1]. Tree reconstruction methods often attempt to take these processes into account, either by using more complex evolutionary models, by removing sites found to violate the assumptions of a simpler evolutionary model, or by those that are predicted to add phylogenetic noise More recent Tree of Life studies have dramatically increased taxonomic sampling and the complexity of the evolutionary models they employ, reconstructing the histories of both Archaea and Bacteria [2,3]. The deep topology of microbial evolution continues to be refined through these efforts, and consensus is emerging for some of the most basal divergences (e.g., the Gracilicutes/Terrabacteria split). Nevertheless, the numerous factors of taxonomic sampling, alignment, and evolutionary models interact in complex ways, and extremely large datasets can make any emergent biases more computationally difficult to test and detect. One technique used to trim sites expected to add phylogenetic noise is the “slow–fast” analysis, whereby the fastest-evolving sites within an alignment are removed to improve the signal-to-noise ratio within the remaining sequence data [4]. In the most extreme cases, all but the slowest-evolving sites may be removed, and the resulting phylogenies are compared (e.g., Raymann et al., 2015 [5]).

The underlying assumption that the slowest-evolving, most-highly conserved sites retain the most accurate phylogenetic signal is largely based on concerns about site saturation—that is, multiple substitutions that overwrite earlier substitutions that would otherwise retain phylogenetic information about deep divergences [6]. Typical saturation assessment methods are based on comparisons between corrected versus non-corrected pairwise distances among sampled taxa [1,6,7]. The problematic nature of fast-evolving sites was initially formulated during a time when maximum parsimony was the major framework of phylogenetic reconstruction [4,6,8], and computing resources constituted a severe bottleneck in modeling evolution. Even considering that substitution saturation can, in general, impede the resolution of phylogenetic relationships between groups, such sites may nevertheless contain informative substitutions for resolving bipartitions of interest.

The likelihood of a site experiencing a substitution along a branch is dependent upon the bipartition branch length and the rate of substitution inferred for that site [9]. Therefore, all things being equal, phylogenetic information for resolving short-branched bipartitions is less likely to be present at slowly evolving sites. Very-slow-evolving sites may not be expected to have experienced any substitutions at all across branches below a certain length threshold, even for a large number of sampled sites. Therefore, the preferential use of slower-evolving sites may be especially problematic if bipartitions of key interest within a phylogeny have short branches. In such cases, slow-evolving sites may be less likely to retain an informative and accurate signal for resolving these bipartitions when compared to faster-evolving sites. This reasoning calls into question the underlying assumption of the “slow–fast” analysis, that, in general, slower evolving sites are more reliable for deeper phylogenetic reconstruction.

Deep Tree of Life phylogenies representing one or more domains of life are often produced from subsets of highly conserved core protein families, such as riboproteins [5,10,11]. Demonstrating that slow- and fast-evolving sites differ in their ability to resolve short-branched bipartitions, this study attempts to evaluate the hypothesis that slow–fast analysis is appropriate for phylogenomic reconstruction of deep species tree relationships. Specifically, given a specific multigene sequence alignment, does trimming fast-evolving sites under the assumption of substitution saturation improve phylogenetic resolution? Conversely, do slow-evolving sites retain more of the informative phylogenetic signal? We observed that removing both the slowest- and fastest-evolving sites from conserved protein alignments should result in improved phylogenetic resolution for the deepest splits in the Tree of Life, specifically those with short branches. Applying this approach to both real and simulated datasets improves the resolution recovered for many deep bipartitions, improving support for the specific evolutionary hypotheses.

## 2. Results

### 2.1. Slow-Evolving Sites Contain Inconsistent Phylogenetic Signals in Conserved Protein Datasets

Sites evolving with different substitution rates are expected to display varying degrees of phylogenetic information, as evidenced by the consistency of the phylogenetic signal. To test this hypothesis, sites from the ribosomal protein alignment published by Hug et al. (2016) were binned into twelve gamma-distributed site-specific substitution rate categories [12], and sites from each bin were used to independently reconstruct the tree. Each of the group sites of the twelve substitution rate categories (SRCs) with similar substitution rates, as evidenced by the most likely substitution rate category identified for the site, were ordered from the slowest- to the fastest-evolving. Invariant sites were omitted before the site rate categories were assigned. Sites from each SRC were used to generate Rate-Specific Alignment Partitions (RSAPs), from RSAP1 (containing the slowest-evolving sites) to RSAP12 (containing the fastest-evolving sites). Each RSAP was used to generate 1000 UltraFast Bootstrap (UFBoot) samples [13] using IQTree [14].

The observed phylogenetic signal differs substantially across RSAPs, as demonstrated by the distinct tree-space areas explored by each UFBoot sample (Figure 1). Surprisingly, in addition to being the most dissimilar to the tree generated from the full alignment, UFBoot samples obtained from RSAP1 also display the most inconsistent topologies among UFBoot replicates (Appendix A), as measured by Robinson–Foulds (RF) distances [15]. The lesser consistency among topologies obtained from slow-evolving sites, when compared to faster-evolving ones, is likely due to low substitution frequencies not generating enough substitutions along branches to resolve many bipartitions. In fact, the similarity between topologies obtained from each RSAP to the whole alignment phylogeny increases with substitution rates up to RSAP9 (average 1.21 substitutions/site). This is followed by a subsequent decrease towards the fastest-evolving RSAPs. The inversion of the positive association between substitution rate and similarity to the topology of the whole alignment after RSAP9 suggests an optimal balance between the accumulated phylogenetic signal and substitution saturation (Figure 1 and Appendix A). Importantly, a higher consistency of phylogenetic signal observed for a given RSAP does not necessarily indicate a greater phylogenetic accuracy, as the true underlying phylogeny remains unknown. However, phylogenetic consistency is a precondition for construction of a well-supported phylogeny, and so serves as a reasonable metric in the absence of a known true tree, even if such a metric does not account for other possible sources of bias.

### 2.2. Short-Branched Bipartitions Are Less Consistently Recovered from Slow-Evolving Sites

Slow-evolving sites are inherently less likely to experience substitutions along short branches than faster-evolving sites; therefore, one would expect slow-evolving sites to less-reliably reconstruct short-branched bipartitions. For example, more than half of bipartitions depicted in Hug et al.’s archaeal subtree have branch lengths shorter than 0.05 substitutions/site, corresponding to 129.8 substitutions among its 2596-site alignment. Since SRC1 possesses an average site-specific substitution rate of 2.332 × 10^−2^ (i.e., accumulating substitutions at a rate 2.332 × 10^−2^ times slower than the average), these sites are expected to accumulate a total of 0.25 substitutions along a 0.05 branch length (0.194% of the total substitutions characterizing the bipartition). Across the same branch length, sites from SRC12 are expected to accumulate a total of 41.36 substitutions (31.86% of the total substitutions characterizing the bipartition) as their average substitution rate is 3.8243 times faster than the average.

Comparing the relative compatibility of bipartitions recovered from each RSAP UFBoot sample to bipartitions found in the tree generated from the full alignment (reference bipartitions), slow-evolving sites are indeed less likely to recover short-branched reference bipartitions. The shorter a bipartition’s branch length, the less likely it is to be consistently recovered by slow-evolving sites (Figure 2A); the faster an RSAP’s substitution rate, the more likely it is to recover shorter-branched bipartitions. There is a significant negative Spearman’s correlation between the average site-specific substitution rate and median branch length of reference bipartitions compatible with UFBoot samples (rho=−0.972 and $p=1.28\times 10^{7}$). We define compatible reference bipartitions as those present in at least 80% of UFBoot samples obtained from a specific RSAP.

### 2.3. Slow-Evolving Sites in Simulated Alignments Are Less Likely to Recover True Tree Bipartitions

Phylogenetic tests of actual sequence datasets can only reliably evaluate consistency across rate categories, rather than accuracy, given that the true underlying tree is inferred rather than known. Therefore, to further assess the accuracy of phylogenetic reconstruction across SRCs, we generated a dataset containing 100 simulated sequence alignments using a known true-tree phylogeny. A random tree topology with 1000 leaves using a branch length distribution modeled after branch lengths observed within the Hug et al. dataset phylogeny was generated and used as a guide tree for sequence simulation (Appendix A). Simulated alignments were then generated by evolving a random starting sequence using the average amino acid composition and substitution frequencies observed in the Hug et al. dataset. As performed for the Hug et al. dataset’s alignment, sites from each simulated alignment were binned according to twelve gamma-distributed site-specific SRCs, generating twelve RSAPs. Each RSAP from each simulated alignment was then used to generate 1000 UFBoot tree samples.

The results of the simulated data analyses were in agreement with those observed for the Hug et al. dataset. RF distances between the true tree and UFBoot samples show that slow-evolving sites consistently underperformed in reconstructing the overall true-tree topology when compared with all other faster-evolving RSAPs (Figure 3). Although substantially less pronounced than in RSAPs from the Hug et al. alignment, the deterioration in phylogenetic signal as substitution rates increase is still present among simulated alignments, detected from RSAP10 onward. The slope of a linear regression between average substitution rate and RF distances between the true tree and UFBoot samples represents the pace in which increasing the former impacts the later, positively or negatively. Among the 100 simulated alignments, the average slope of the described regression from SRC9 to SRC12 is 26.23 (Appendix A), suggesting that although deviation from the true tree still increases in the simulated scenario, it does so 45 times slower than that observed among equivalent SRCs from the Hug et al. [10] dataset (Appendix A).

### 2.4. Slow-Evolving Sites in Simulated Sequence Datasets Are Biased against Reconstructing True Short-Branched Bipartitions

A further breakdown of true bipartition recovery frequencies from simulated RSAPs shows that these are not independent of site rates. A total of 91 out of the 100 simulated alignments showed significant negative Spearman’s correlations between the average substitution rate and the branch length of bipartitions present in at least 80% of UFBoot samples ($\bar{rho}=-0.69$ and q<0.05). True tree bipartitions with shorter branch lengths were more frequently reconstructed by faster-evolving than by slower-evolving sites, as shown by the fastest- and slowest-evolving RSAPs reconstructing 73% and 18% of bipartitions with branch lengths below the median, respectively (Figure 2B). Interestingly, there was no detected correlation between reconstructing true bipartitions with long branches across substitution rate categories. Remarkably, UFBoot samples from all simulated RSAPs were consistent in accurately recovering ~99% of the top-25% longest-branched bipartitions. This suggests that (1) given the large absolute number of substitutions present in such bipartitions, they can be found even among the slowest-evolving sites; and (2) substitution saturation in fast-evolving sites does not necessarily significantly obscure the phylogenetic signal, or impair phylogenetic reconstruction, even along long branches where many such substitutions are expected to occur.

### 2.5. Substitution Saturation Does Not Explain the Loss of Phylogenetic Signal from Fast-Evolving Sites

The minor deterioration in phylogenetic signal among the fastest-evolving sites from simulated alignments is unexpected given the high degree of substitution saturation present in simulated alignments (Appendix A), especially when compared to the much-more-dramatic deterioration observed for the Hug et al. dataset. Since branch lengths and substitution rate categories from simulated datasets were modeled after the Hug et al. dataset, saturation by itself cannot be the major force driving the deterioration in phylogenetic signal among the fastest-evolving sites in the real sequence data. The concept of substitution saturation causing a loss of the phylogenetic signal has been questioned before using small datasets and limited taxonomic sampling [8]. In other words, these results suggest that the consistent deterioration in phylogenetic signal among the fastest-evolving sites is more likely to be due factors not replicated in the simulated dataset. These potentially include alignment errors [16] and/or the fitting of a sub-optimal substitution model [17,18,19]. These two possibilities were further investigated.

Regardless of the phylogenetic information contained in an aligned site, fast-evolving sites are more prone to misalignment simply due to the increased number of states shared by a subset of taxa at nonhomologous sites [16]. As both ancestral relationships between taxa and their internal sequence states are unknown variables in phylogenetic reconstruction, it is impossible to directly assess alignment accuracy of empirical datasets. However, given the greater diversity of alignment variations possible within gap-rich regions, sites flanked by gap-rich regions should be enriched in misaligned residues. As such, these were used as a proxy for poorly aligned sites. Conversely, sites within blocks of sequence with few gaps are less likely to be misaligned given that there are fewer combinations of similarly scored local alignment variations. By restricting the sampling of fast-evolving sites (from SRC10 to SRC12) exclusively to sites flanked by ungapped regions of the alignment, the resulting RSAPs are expected to contain substantially fewer misaligned sites, and therefore more closely match the profile observed for simulated sites evolving at similar rates. Indeed, the generated UFBoot samples from these RSAPs became significantly more similar to the reference tree (Figure 4). Furthermore, as expected, restricting fast-evolving site sampling to sites flanked by gap-rich regions resulted in UFBoot sample phylogenies that were even more dissimilar to the reference tree. This result supports that misalignment plays a significant role in the loss of phylogenetic accuracy among fast-evolving sites, independent of saturation effects.

The impact of sub-optimal substitution models was also assessed. Trees were generated from simulated alignments using a model that was different to that which the alignments were generated from, specifically, the Dayhoff model [20] instead of LG [21]. Sites from simulated alignments were binned into SRCs using the Dayhoff substitution model, followed by RSAP construction, and UFBoot replicates were also generated under the Dayhoff model. UFBoot replicates generated under the Dayhoff model showed a significant increase in the average slope of linear regressions (q<0.05, Appendix A), suggesting that fast-evolving sites are more sensitive to poorly fitted substitution models, contributing to an inaccuracy of phylogenetic reconstruction. Interestingly, UFBoot replicates from slow-evolving sites (SRC1 to SRC3) reconstructed under the Dayhoff model were more similar to the true tree than those reconstructed under the “true” LG model (Appendix A). These results might be explained by the cumulative impact of substitution probabilities. The more substitutions a site experiences along its history, the more impactful small changes in the model tend to be, which also increases errors related to overfitting, as more events are fitted into a single site [17,18,19].

Based on these results, we conclude that substitution saturation alone is not the main cause of phylogenetic inaccuracy observed for fast-evolving sites. Rather, it is likely that an increased frequency of indels amongst the fastest-evolving sites leads to misalignment in some sequence regions, and to poor model fitting having a greater impact. Both of these processes may therefore obscure the potential for these sites to reconstruct phylogenetic information despite sequence saturation effects.

### 2.6. Phylogenetic Reconstructions Using Rate-Specific Subsets of Sequence Alignment Data

The relatively short internal branch lengths of bipartitions in the phylogeny reported by Hug et al. (e.g., median branch length of 0.05 substitutions/site) is likely related to the general underperformance of slow-evolving sites in recovering this topology. The results from simulated alignments suggest that objective alignment trimming strategies based on site-specific substitution rate categories could improve phylogenetic reconstruction from this dataset, as well as other comparable datasets. In order to test the impact of these strategies, three distinct site trimming approaches were used in reconstructing the phylogeny of the archaeal and eukaryal subtree from the Hug et al. dataset: (1) trimming slow-evolving sites only, from SRC1 to SRC4 (Figure 5B,F,J); (2) trimming fast-evolving sites only, from SRC10 to SRC12 (Figure 5C,G,K); and (3) trimming both slow- and fast-evolving sites (Figure 5D,H,L). Alignment partitions resulting from trimming either slow- or fast-evolving sites contained 1886 and 2130 sites, respectively (72.6% and 82% of the whole alignment, respectively). However, trimming both slow- and fast-evolving sites led to an alignment partition with only 1442 sites binned from SRC5 to SRC9, corresponding to 55% of the whole alignment (Appendix A). Alignments resulting from each trimming strategy, together with the whole alignment, were then used for phylogenetic reconstruction under three distinct substitution models: LG+G, the C60 mixture model, and the distribution free LG4X. Major relationships within the archaeal/eukaryal tree were significantly impacted by these test conditions (Figure 5). For example, the grouping of Asgardarchaeota with Eukarya was recovered in only five of the assessed combinations: the whole alignment under any substitution model (Figure 5A,E,I), trimming fast-evolving sites under the LG+G model (Figure 5C), and by trimming both slow- and fast-evolving sites under the C60 mixture model (Figure 5H), although this was not highly supported in any case. The remainder of the reconstructed phylogenies, seven out of twelve, presented Eukarya as a sister to the TACK superphylum, and Asgardarchaeota as a sister to the Eukarya+TACK clade, a topology which has also been recently proposed using a distinct dataset [22]. The only combination of alignment and tree reconstruction model to recover a well-supported (i.e., UFBoot > 95%) placement of Eukarya was found by trimming both slow- and fast-evolving sites under the LG+G model. In this case, the TACK superphylum is a sister to Eukarya (Figure 5D).

The combination of short internal branch lengths within the Archaea+Eukarya clade with the extremely long branch leading to Bacteria, and the relatively small number of concatenated aligned proteins (16 ribosomal proteins) prevented the realistic expectation that this dataset would recover an accurate rooting for Archaea+Eukarya [22,23]. Nevertheless, recovered rootings to terminal branches remain more suspect, as they are more likely to represent long branch attraction artifacts in the absence of a true phylogenetic signal [24]. Phylogenies reconstructed from alignment partitions with both slow- and fast-evolving sites trimmed led to deeper Archaea+Eukarya roots when reconstructed under LG+G or LG4X substitution models (Figure 5D,L). Under the C60 substitution model this alignment partition resulted in a somewhat shallower rooting (Figure 5H) when compared to the whole alignment (Figure 5E), although this still recovered a deeper root than from trimming either the faster or slower sites (Figure 5F,G).

### 2.7. Phylogenetic Impact of Fast-Evolving Sites within Gap-Rich Regions

In the light of previously discussed results (i.e., substitution saturation not leading to a deterioration in phylogenetic signal among fast-evolving sites), we reconstructed the Hug et al. dataset phylogeny by trimming only fast-evolving sites flanked by gap-rich sites from the whole alignment. The 249 sites fulfilling both requirements, being fast-evolving and within gap-rich regions, were trimmed from the whole alignment and the resulting alignment partition submitted for reconstruction under three evolutionary models were: LG+G, C60, and LG4X (Figure 6). Reconstructions using all three substitution models reported the TACK superphylum as a sister to Eukarya, with high bipartition support (93% UFBoot support) under the LG+G model (Figure 6A). Asgardarcheota was reported as being a sister to the Eukarya+TACK clade by all three reconstructions.

### 2.8. Composition Heterogeneity among Substitution Rate Categories

Differences in phylogenetic consistency observed across RSAPs may also be driven by site rate-specific compositional biases, especially when such biases violate assumptions of substitution models. Sites evolving at varying rates may adapt to changes in the underlying mutation bias with varying efficiencies [25]. The slowest-evolving sites do indeed show substantial compositional bias when compared to the whole alignment (Table 1). Such compositional bias is not uniformly distributed among all twenty amino acids, as highlighted by the massive enrichment of glycine within RSAP1 sites (Appendix A). The distinct biases detected among RSAPs lead to their amino acid frequencies being better modeled by distinct combinations of substitution probabilities (Dayhoff, JTT, WAG, and LG) and amino acid frequencies (default and empirical) when assessed using homogeneous substitution rates (Table 1).

While empirical amino acid frequencies may represent a better overall fit with alignment data in some cases, if the bias is largely provided by the least informative sites, adopting empirical frequencies may not be the optimal approach. Applying empirical or default amino acid frequencies from alignments with either slow- or fast-evolving sites trimmed produced substantial changes in the reconstructed topology, as measured by the RF distances between the UFBoot samples (Appendix A). Trimming both slow- and fast-evolving sites reconstructed topologies that were significantly more robust to amino acid frequency changes (Appendix A) despite these alignments containing fewer sites. When jackknifed to the same alignment length as the partition with trimmed slow- and fast-evolving sites, eight out of ten jackknives from the whole alignment were shown to be more sensitive to amino acid frequency changes than the alignment partition with trimmed slow- and fast-evolving sites (Appendix A). Using default or empirical amino acid frequencies with the full alignment substantially impacts phylogenetic reconstruction for the dataset, recovering either 2-Domain or 3-Domain topologies, respectively (Figure 7).

### 2.9. Short Deep Branches Increase in Frequency with Increased Taxon Sampling

Branch lengths are representations of evolutionary distances between two sequence states; in the case of internal bipartitions, this distance is effectively the distance between two inferred ancestors of the sampled taxa. A bipartition with a long branch length is often a consequence of poor sampling within a group [26]. This may be caused by: patterns of extinction, in which case intermediates do not exist to be sampled; the inability to sample unknown or unsequenced lineages; or the deliberate down-sampling of taxa for tractable phylogenetic analysis or taxon sampling balance. Improving taxonomic sampling within a group consequently increases the number of reconstructed intermediary ancestors (nodes) within it. The more intermediary nodes reconstructed between taxa, the more closely related these will be, leading to shorter branch lengths in the reconstructed phylogeny.

Given the performance of fast-evolving sites in recovering short-branched bipartitions, the utility of fast-evolving sites in resolving phylogenies should therefore increase as more genomes are sequenced and coverage of genomic diversity becomes denser, as this increases the predominance of short-branch bipartitions in phylogenies. To illustrate this, branch lengths were re-estimated using random subsamples of the Hug et al. dataset. A clear downward trend in branch length was observed as taxon sampling increased. Sample sizes were gradually increased from 10% up to 90%, each replicated 100 times (Figure 8). The negative correlation between branch length and sample size emphasizes the importance of better characterizing the impact of rate-specific site partitioning in phylogenetic reconstruction as taxonomic coverage continues to increase.

## 3. Conclusions

It is common practice among microbial phylogenetic studies to trim fast-evolving sites based on substitution saturation metrics obtained using slow–fast analysis [5,27,28]. Testing the assumptions underlying this methodology using both real and simulated datasets based on ribosomal protein alignments and Tree-of-Life-level sequence diversity, it appears that fast-evolving sites overperform slow ones with respect to the consistency and accuracy of phylogenetic reconstruction. The poorer performance of slow-evolving sites is especially apparent in reconstructing bipartitions with short branch lengths. When compared to sites from the middle of the substitution rate spectrum, fast-evolving sites do show a significant loss of accuracy during phylogenetic reconstruction using both empirical and simulated sequence alignment datasets. However, comparisons using empirical and simulated datasets show that misalignment and poor model specification are more likely to negatively affect phylogenetic accuracy among fast-evolving sites than substitution saturation, corroborating previous theoretical findings based on less comprehensive datasets [8].

Results obtained using the Hug et al. dataset show that the phylogenetic signal present within slow-evolving sites is less consistent than that found within fast-evolving sites, especially for short-branched bipartitions. This observation is corroborated by analyses on simulated sequence alignment data, using a predetermined phylogeny and a known evolutionary model. This strongly suggests that the reported results are not artifacts due to the unaccounted for evolutionary processes, misalignment, or model biases. Among the simulated alignments, the fastest-evolving sites, expected to have experienced substitution saturation, still provided more accurate phylogenetic reconstructions than slow-evolving sites. The underperformance of highly conserved sites is accentuated in large-scale phylogenies as bipartition branch lengths tend towards fewer substitutions per site [5,10,28]. It follows that, as taxonomic sampling continues to increase, faster-evolving sites become more necessary for reliable phylogenetic reconstruction. These results challenge the assumption that the slower a site accumulates substitutions, provided it is not invariable, the better it is suited to reconstructing deep phylogenies [4,29]. Subjective alignment trimming strategies can generate phylogenies with significant differences in key bipartitions and can also impact the extant phylogenetic signal by affecting estimated model parameters (Table 1). The discussed results are not aimed towards proposing a new Tree of Life or evaluating any specific Tree of Life hypotheses, but rather, they test the merits of current alignment trimming strategies and demonstrate how subjectively trimming fast-evolving sites may impair progress in these endeavors.

## 4. Materials and Methods

### 4.1. Hug et al. [10] Dataset

Hug et al. [10] published a 3083-taxa two-domain Tree of Life, recovering Eukarya as a sister to Asgardarchaeota, which are both within the TACK superphylum. The tree was reconstructed from a 2596-site super-matrix resulting from concatenating 16 ribosomal proteins, without the trimming of fast-evolving sites or gene partitioning. The multiple sequence alignment resulting from the concatenated 16 ribosomal proteins was obtained from Appendix A, available as part of the Hug et al. [10] publication.

Aligned site positions were classified into twelve gamma-distributed substitution rate categories (α=0.803) using IQTree’s parameter “-wsr” [12]. Sites from each SRC were repeated in tandem to obtain the same length as the published alignment so that every RSAP has the same number of informative sites. RSAP phylogenies were reconstructed using IQTree with LG+G, and pairwise distances between tree topologies were measured using Robinson–Foulds distances, as implemented in IQTree.

During the analysis of different alignment partitions to reconstruct phylogenies without slow- and/or fast-evolving sites, we defined sites from SRC1 to SRC4 as slow, and sites from SRC10 to SRC12 as fast.

### 4.2. Sequence Simulation

Simulated sequence alignments containing 1000 taxa were generated using Indelible software [30]. A random tree topology was generated using the ETE Toolkit [31], and internal and terminal branch lengths were independently assigned from a gamma-distributed branch length obtained from the Hug et al. phylogeny. Shape parameters of gamma-distributed internal and terminal branch lengths were 0.7581720 and 1.509421, respectively. Sequence simulations were performed with 100 replicates under an LG model with 12 gamma-distributed site-specific substitution rate categories, no invariant sites, no indels, and the same amino acid frequency as the Hug et al. dataset (control file available in Appendix A).

### 4.3. Phylogenetic Analysis

All phylogenies were reconstructed using IQTree [14], and when necessary, automatic model selection was performed using the “-m MFP” parameter. Ultrafast Bootstrap samples were obtained using the UFBoot method [13] and pairwise Robinson–Foulds distances [15] between trees were both estimated using IQTree.

RSAPs were generated by replicating sites from its respective SRC until the same length of the whole alignment was obtained. This was carried out to normalize the number of informative sites present in different SRCs.

## Figures and Tables

**Figure 1 microorganisms-11-02499-f001:**
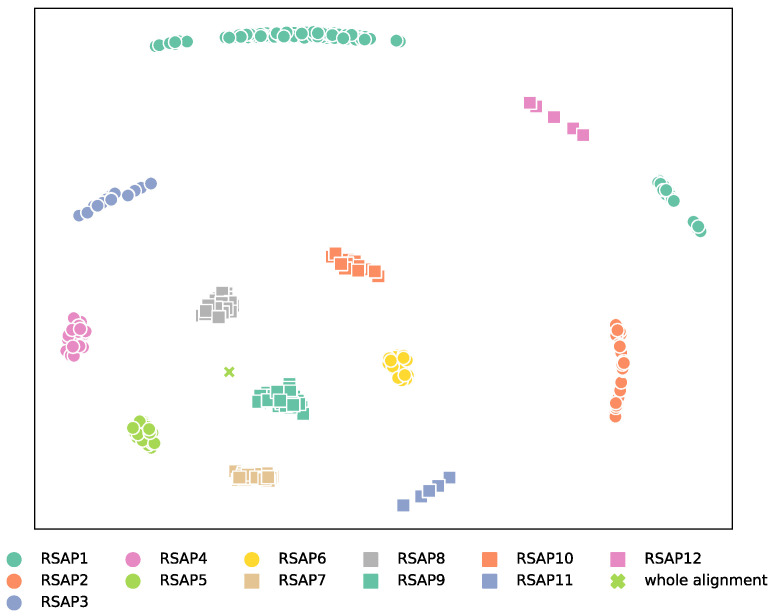
Multidimensional scaling of RF distances between UFBoot samples and reference topologies. The relative distances between tree topologies within each RSAP informs how sampled trees obtained from a RSAP are consistently similar (e.g., RSAP8 and RSAP9) or dissimilar (RSAP1 and RSAP12) to the overall topology. One hundred trees were randomly chosen from each UFBoot sample. The RF distances between each set of trees, as well as the phylogeny published by Hug et al. [10] was then calculated.

**Figure 2 microorganisms-11-02499-f002:**
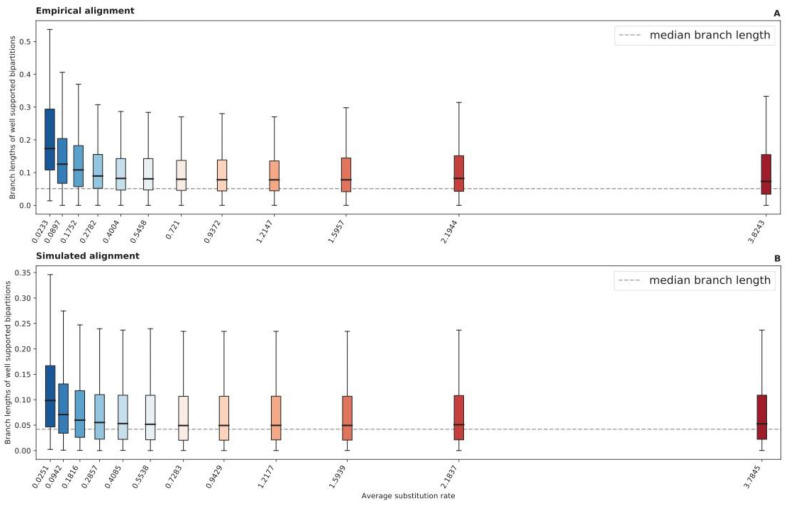
Boxplot distributions of reference bipartitions present in at least 80% of UFBoot samples generated from each RSAP, for both Hug et al. (**A**) and simulated (**B**) datasets. The dashed line represents the median internal branch length of each reference phylogeny.

**Figure 3 microorganisms-11-02499-f003:**
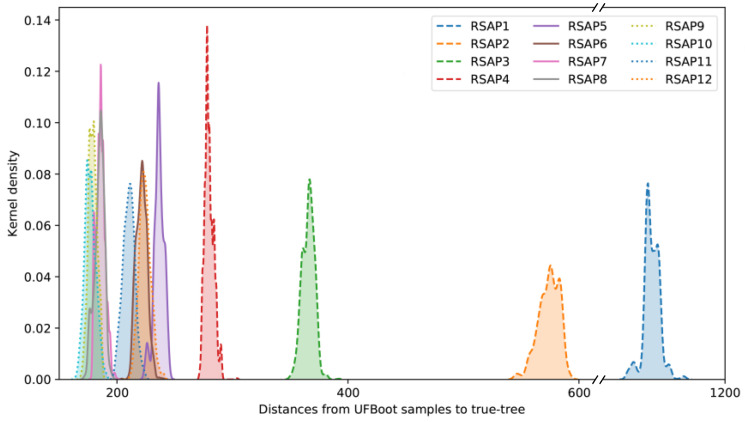
Distribution of RF distances between RSAP UFBoot samples and the true tree topology. Distances plotted are all derived from a single simulated alignment replicate, as distributions are consistent across all 100 replicates.

**Figure 4 microorganisms-11-02499-f004:**
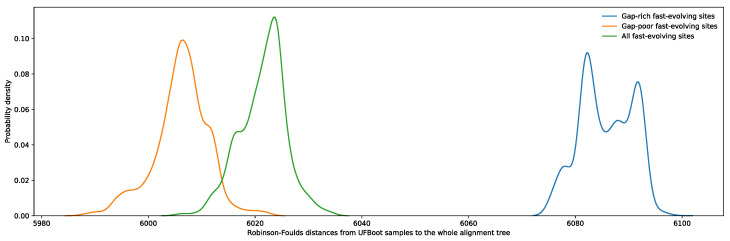
Distribution of Robinson–Foulds distances between the phylogeny proposed by Hug et al. and UFBoot samples obtained using three subsets of fast-evolving sites, SRC10 to SRC12: (1) fast-evolving sites flanked by any number of gaps in green, (2) fast-evolving sites flanked by gap-rich regions in blue, and (3) fast-evolving sites flanked by gap-poor regions in orange. Gap-rich regions were defined as containing 2000 gaps or more, while gap-poor regions were defined as containing 500 or less gaps. A total of 105 sites from distributions 1 and 2 were randomly selected and used to generate UFBoot samples in order to match the number of fast-evolving sites flanked by gap-rich regions.

**Figure 5 microorganisms-11-02499-f005:**
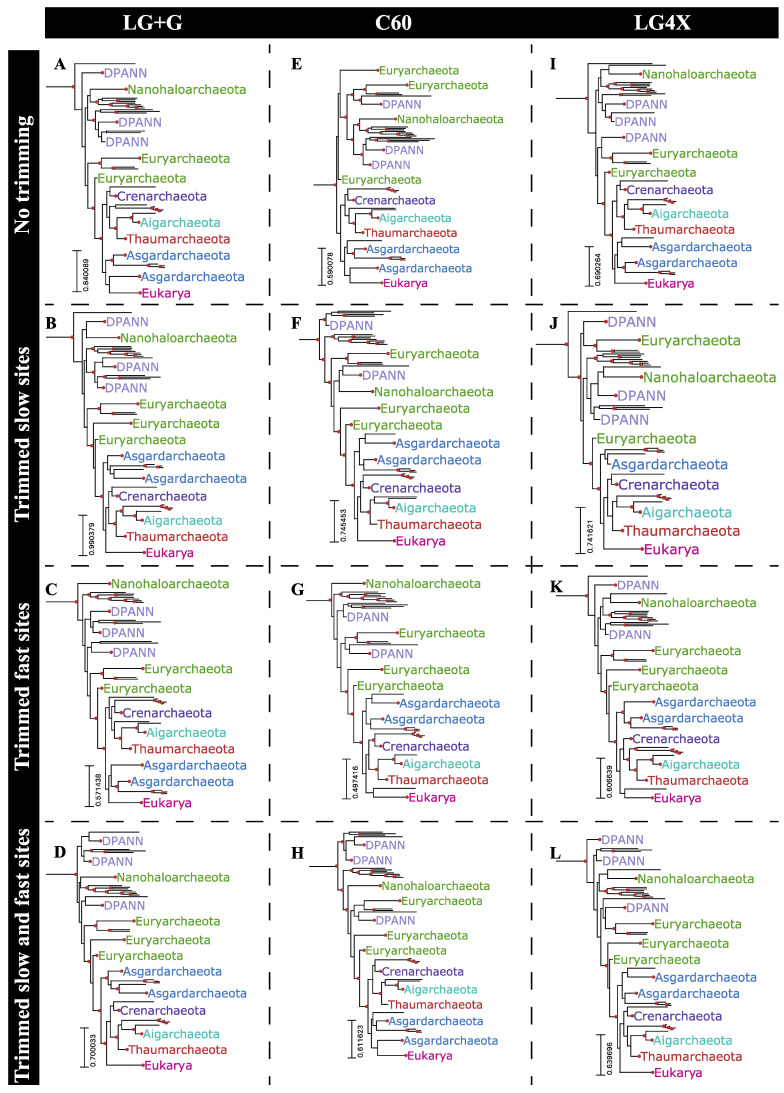
Archaea+Eukaryota subtrees obtained by submitting four distinct alignment partitions of the Hug et al. [10] dataset (whole alignment: (**A**,**E**,**I**); trimmed slow-evolving sites: (**B**,**F**,**J**); trimmed fast-evolving site: (**C**,**G**,**K**); and both trimmed slow- and fast-evolving sites: (**D**,**H**,**L**)) to three distinct substitution models (LG+G: (**A**–**D**); C60: (**E**–**H**); and LG4X: (**I**–**L**)). Highlighted bipartitions have a UFBoot support greater than 95%.

**Figure 6 microorganisms-11-02499-f006:**
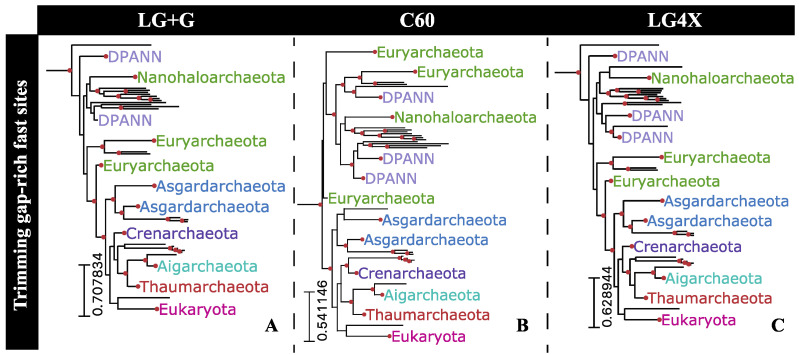
Eukarya+Archaea subtrees reconstructed by removing 249 fast-evolving sites flanked by gap-rich sites from the Hug et al. [10] dataset. Topologies were reconstructed under three distinct substitution models: (**A**) LG+G, (**B**) C60, and (**C**) LG4X. Highlighted bipartitions have a UFBoot support greater than 95%.

**Figure 7 microorganisms-11-02499-f007:**
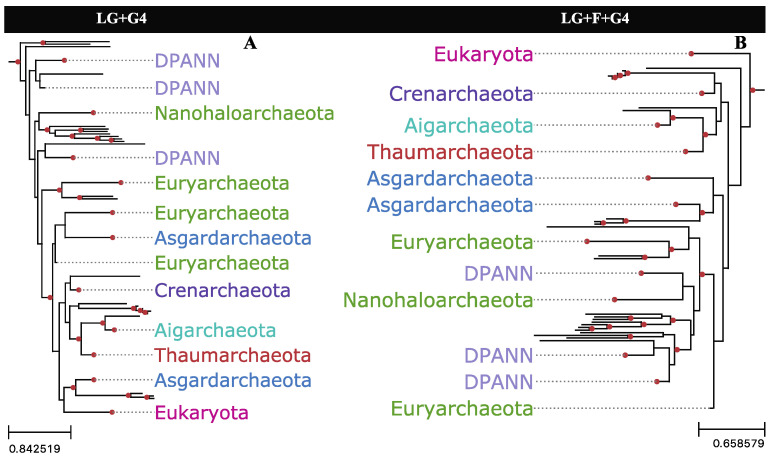
Archaea+Eukarya subtrees reconstructed from the full Hug et al. [10] alignment, with varying amino acid frequency parameters in the substitution model. (**A**) Tree reconstructed with equilibrium frequencies defaulting to the LG substitution model; (**B**) Tree reconstructed with empirical frequencies as estimated from the alignment. Nodes with UFBoot support of 95% or greater are highlighted by red circles.

**Figure 8 microorganisms-11-02499-f008:**
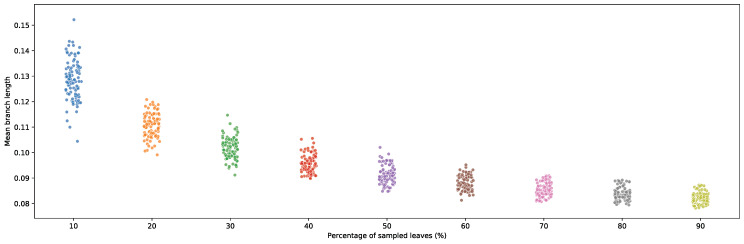
Average re-estimated branch lengths from 100 random taxonomic subsamples of leaves present in the Hug et al. [10] published phylogeny. Subsampling was increased by 10% at each step.

**Table 1 microorganisms-11-02499-t001:** Best-fitting substitution models of each RSAP from the Hug et al. [10] dataset.

RSAP	1	2	3	4	5	6	7	8	9	10	11	12
**Average site-specific substitution rate**	0.023	0.089	0.175	0.278	0.4	0.545	0.721	0.937	1.214	1.595	2.194	3.824
**Sum of squared errors of aa compositional bias**	0.14	0.012	0.003	0.002	0.003	0.003	0.003	0.006	0.005	0.009	0.01	0.01
**Best-fit substitution model**	WAG	LG	LG	LG+F	LG+F	LG+F	LG	LG	LG	WAG	WAG	WAG

## Data Availability

The data underlying this article are available in the article and in its online Appendix A. INDELible’s configuration file required to reproduce simulated alignments, as well as its underlying tree, used in this study are available in the Appendix A. Alignments resulting from SRCs combinations (i.e., removed slow-evolving sites, removed fast-evolving sites, and removed both slow- and fast-evolving sites) as well as their resulting phylogenies are available in Appendix A. Simulated alignments obtained from INDELible are available in Appendix A (https://doi.org/10.6084/m9.figshare.14126768.v1, accessed on 29 September 2023). The empirical alignments used in this study are available in the Appendix A of their respective publications [10].

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
