# Peer review of "Fast-Evolving Alignment Sites Are Highly Informative for Reconstructions of Deep Tree of Life Phylogenies"

_microorganisms, 2023, doi:10.3390/microorganisms11102499_

Round 1
Reviewer 1 Report
Rangel and Fournier present a nice examination of the biases involved in analyzing only some rate categories in mutli-sequence alignments. This is an important area of study, and I'm glad the authors have done this analysis. Overall I would say this is a useful contribution, and I have a few comments that may help improve the manuscript.
-Figure 8 does not show up on my PDF.
-I would appreciate more background on other recent Tree of Life studies. Right now the references are rather short, and I think some added context in the intro could be helpful. The hug et al topology differs quite a bit from more recent studies, so perhaps pointing that out would be useful context.
-The authors use the Hug et al 2016 study, which has a number of known errors. It is now known that the Patescibacteria are a sister group to the Chloroflexi, not basal-branching as shown there. Do any of the partitions in this study recover the correct placement of the Patescibacteria? That could be one useful piece of information that would guide us towards the most useful rate category to use for these analyses.
Author Response
Rangel and Fournier present a nice examination of the biases involved in analyzing only some rate categories in multi-sequence alignments. This is an important area of study, and I'm glad the authors have done this analysis. Overall I would say this is a useful contribution, and I have a few comments that may help improve the manuscript.
-Figure 8 does not show up on my PDF.
We are sorry for this problem when exporting to PDF, we have fixed it.
-I would appreciate more background on other recent Tree of Life studies. Right now the references are rather short, and I think some added context in the intro could be helpful. The hug et al topology differs quite a bit from more recent studies, so perhaps pointing that out would be useful context.
The reviewer is correct that many more extensive Tree of Life (ToL) phylogenies have been recently published, and some aspects of the inferred topology of the Tree of Life has been revised based on this work. While our work is not meant to be a review of the current state of ToL studies, or to improve on any of the specific conclusions therein, we do appreciate the suggestion that additional context would help the reader. We now briefly comment on these studies in the introduction section (see below). The results and conclusions of our work presented here are independent of any particular tree topology, and, rather, indicate how sensitive the robustness of these inferences are to sampling based on site rate categories. The simulation analyses we report here also control for many of the methodological errors which may occur during the analysis of real data, showing that these conclusions are generalizable, and are not a quirk of the Hug dataset.
“More recent Tree of Life studies have dramatically increased taxonomic sampling and the complexity of the evolutionary models they employ, reconstructing the histories of both Archaea and Bacteria [2,3]. The deep topology of microbial evolution continues to be refined through these efforts, and consensus is emerging for some of the most basal divergences (e.g., the Gracilicutes/Terrabacteria split). Nevertheless, the numerous factors of taxonomic sampling, alignment, and evolutionary models interact in complex ways, and extremely large datasets can make any emergent biases more computationally difficult to test and detect.”
-The authors use the Hug et al 2016 study, which has a number of known errors. It is now known that the Patescibacteria are a sister group to the Chloroflexi, not basal-branching as shown there. Do any of the partitions in this study recover the correct placement of the Patescibacteria? That could be one useful piece of information that would guide us towards the most useful rate category to use for these analyses.
Thank you for raising this point, which ideally would be one of the hypothesis this approach is helpful with. In this case specifically, none of the substitution rate combinations returned the “correct” answer. In all surveyed scenarios (i.e., trimmed fast-evolving sites, trimmed slow-evolving sites, and trimmed both fast- and slow-evolving sites) the resulting phylogeny places Patescibacteria basally. Given this dataset, this is likely an artefact of the taxon and sequence sampling driving this specific topology.
Reviewer 2 Report
The authors of study "Fast-evolving alignment sites are highly informative for reconstructions of deep Tree of Life phylogenies" present a clear and well-presented manuscript. it presents a phylogenetic analysis of Fast-evolving regions that can be applied in various large-scale phylogenetic analyses. however, some methodological aspects need to be clarified.
Main highlight
A) Why didn't the authors consider making a comparison between genes or regions with intermediate evolution with the fast evolving data? Such a comparison could better support the results presented
B) The database or alignments used for the analyses should be included in the supplementary materials. there is no way to compare the results.
C) There is no discussion of results section. this is an important part of the research. why?
Minor highlight
1) In the line 60, it is suggested to address the problem of the origin of polytomies in a phylogenetic tree. Could this also be resolved with rapid evolution sites?
2) In lines 65 to 86, results and conclusions are presented in the introduction section. These should be replaced by a hypothesis and a general objective.
3) In lines 89 to 100, this information is methodological and not results. please edit it in the appropriate section.
4) In the line 432, the genes and the order of the 16 ribosomal proteins concatenated and the bioinformatics program should be indicated.
5) In the line 433, the alignment program used must be included.
Author Response
The authors of study "Fast-evolving alignment sites are highly informative for reconstructions of deep Tree of Life phylogenies" present a clear and well-presented manuscript. it presents a phylogenetic analysis of Fast-evolving regions that can be applied in various large-scale phylogenetic analyses. however, some methodological aspects need to be clarified.
Main highlight
- A) Why didn't the authors consider making a comparison between genes or regions with intermediate evolution with the fast evolving data? Such a comparison could better support the results presented.
As the scope of our study is to assess the impact of substitution-rate based site trimming, re-generating the Hug et al. phylogeny with a different set of genes would defeat the purpose of direct comparison. That said, we did compare the results obtained when trimming both slow- and fast-evolving sites (i.e., only using intermediate substitution-rate sites) as shown in Figure 5, sub panels D, H, and L. In line 311 we have discussed the following finding:
“Phylogenies reconstructed from alignment partitions with both slow and fast-evolving sites trimmed led to deeper Archaea+Eukarya roots when reconstructed under LG+G or LG4X substitution models (Figures 5d and 5l). Under the C60 substitution model this alignment partition resulted in a somewhat shallower rooting (Figure 5h) when compared to the whole alignment (Figure 5e), although still recovering a deeper root than from trimming either the faster or slower sites (Figures 5f and 5g)” – lines 311:316
- B) The database or alignments used for the analyses should be included in the supplementary materials. there is no way to compare the results.
Thank you for pointing this out, the datasets are available at FigShare (https://doi.org/10.6084/m9.figshare.14126768.v1) but its mention was not added to the current iteration of the manuscript. We have corrected it and the Supplementary Data is now referenced in the “Data Availability Statement”.
- C) There is no discussion of results section. this is an important part of the research. why?
Given the large number of individual analyses constituting the work, we decided to discuss each set of results in turn, rather than all at once in a separate discussion section. We believe this improves the readability of the manuscript.
1) In the line 60, it is suggested to address the problem of the origin of polytomies in a phylogenetic tree. Could this also be resolved with rapid evolution sites?
Theoretically, including fast-evolving sites is more likely to yield resolved bipartitions since they have undergone more substitutions. That being said, our study is based on maximum likelihood reconstructions, which do not yield polytomic bipartitions. Even though we have briefly touched on bipartition support during the discussion, collapsing low-support bipartitions into polytomies is part of downstream processing of the phylogenetic reconstruction, and the discussion of the impact of substitution-rate trimming strategies to the multiple approaches to estimate bipartition support is outside of the scope of our manuscript.
2) In lines 65 to 86, results and conclusions are presented in the introduction section. These should be replaced by a hypothesis and a general objective.
We appreciate the reviewer’s feedback, and have edited the last Introduction paragraph accordingly (see below).
“Deep Tree of Life phylogenies representing one or more Domains of life are often produced from subsets of highly conserved core protein families, such as riboproteins (3, 8, 9). Demonstrating that slow- and fast-evolving sites differ in their ability to resolve short branched bipartitions, this work attempts to evaluate the hypothesis that slow-fast analysis is appropriate for phylogenomic reconstruction of deep species tree relationships. Specifically, given a specific multigene sequence alignment, does trimming fast-evolving sites under the assumption of substitution saturation improve phylogenetic resolution? Conversely, do slow-evolving sites retain more informative phylogenetic signal? We observe that removing both the slowest and fastest evolving sites from conserved protein alignments should result in improved phylogenetic resolution for the deepest splits in the Tree of Life, specifically those with short branches. Applying this approach to both real and simulated datasets improve the resolution recovered for many deep bipartitions, improving support for specific evolutionary hypotheses.” – lines 70:82
3) In lines 89 to 100, this information is methodological and not results. please edit it in the appropriate section.
We appreciate the reviewer’s feedback and agree that the first paragraph of the Results section might be more technical than normally expected in the section. That said, given the manuscript organization we have used (i.e., Methods as the last section in the manuscript) we believe that a short contextualization of the methods used for the described results improve the readers comprehension.
4) In the line 432, the genes and the order of the 16 ribosomal proteins concatenated and the bioinformatics program should be indicated.
Multiple sequence alignment of ribosomal genes were obtained from Supplementary Dataset 1 of the Hug et al. (2016) publication. The original alignment was not re-calculated. We have added extra information to the text to make it clearer.
“The multiple sequence alignment resulting from the concatenated 16 ribosomal proteins was obtained from Supplementary Dataset 1 available as part of the Hug et al. (2016) publication” – lines 430:432
5) In the line 433, the alignment program used must be included.
Same as above, sites from the original publication were binned according to relative substitution rate categories. Multiple sequence alignment was not re-calculated.
I have edited this section to focus only on the key hypotheses and tests made, and to fix some of the language. Check and then replace the manuscript text with the revised text.
Round 2
Reviewer 2 Report
I wish to thank Author for their efforts to adjust the manuscript following my suggestions